# Energy Bidding Quadratic Model and the Use of the B-Loss Matrix for Determining Consumer Energy Price

**Jangkung Raharjo [1,\*] and Hermagasantos Zein [2]**

1   School of Electrical Engineering, Telkom University, Bandung 40257, Indonesia
2   Energy Conversion Engineering Department, State of Polytechnic of Bandung, Bandung 40559, Indonesia
\*   Correspondence: jangkungraharjo@telkomuniversity.ac.id; Tel.: +62-2287885901

**Abstract:** The liberalization trend has led to electric restructuring in market industries. At the start of the 1990s, it was recommended to shift the electricity business from a monopoly to a competitive system. The electric power problem becomes more complex from competition because competitors must be ready to win or lose. The method that has succeeded in determining energy prices in competition is the locational marginal price method implemented by the New York Service Operator. In general, the characteristic of the supplier offers are in step function forms, so optimizing prices and allocating transmission losses are a problem. This paper proposes a method for determining electrical energy prices on the consumer side in each location. The method uses a quadratic approach to perform direct method optimization. The transmission losses are calculated through the B-loss matrix approach, and then allocations of the transmission losses are separated with the proportional method. Simulation results for three locations with six suppliers, as well as on a larger scale (118 buses, 54 generators) were obtained.

**Keywords:** industrial liberalization; competition winners; energy prices; quadratic approach; B-loss matrix

## 1. Introduction

The energy crisis is a serious phenomenon that is of concern now and in the future. Various efforts continue to be taken to improve the efficiency of power generation. This efficiency problem has experienced its highest peak since the application of a generator with a combination cycle with a high level of efficiency. However, the threat of lack of energy continues until this day, caused by the reduced fossil reserves, while the demand growth rises sharply. As a result, the energy price of fossils in the future will be tough to estimate, although renewable energy sources have been developed in the last two decades. In the early 1990s, experts focused on energy savings, operational efficiency, and transparency. The results are a message for changing the electricity business to a market system as a competition system [1].

In an effort to achieve prosperity for electricity users, competition efforts to create healthy electricity trading conditions have been carried out [2], and one of the indicators of success in the competition is shown by the optimal selling price for consumers. However, to achieve competitive selling prices, power plants must undertake various strategies to further improve their production efficiency [3]. An important thing that also plays a role in influencing electricity costs is the transmission network that delivers the electricity to the customer. Compared to an integrated system, the roles of the network in a competition system will be complex because it is open to all market participants [4]. In 1990, the UK restructured the power supply industry. As an implication of the restructuring, several private companies are competing to contribute significantly to the electricity supply for the next period. Therefore, a new term Main Electricity Producers (MPPs) is introduced for designated companies whose main purpose is to generate electricity [5].

The success of electricity trading is shown by the following facts, such as Pennsylvania-New Jersey-Maryland (PJM) [6], Competition in New York: NYISO [7], in New England: ISO-NE [8], in California: CAISO [9], in Texas: ERCOT [10], Midcontinent Independent System Operator [11], and Neidhardt Engineering and Manufacturing. The success of those markets is due to the implementation of locational marginal price (LMP), while other markets are heading toward LMP. LMP is the essential key in evaluating the electric energy market, and GENCO has a large share in the power market [12]. The LMP mechanism was first discovered by [13] and was introduced in PJM-ISO. However, the basis of the LMP mechanism is the determination theory of spot pricing proposed by [14]. The hallmark of the LMP mechanism is that all scheduling generators from suppliers (competition participants and bilateral transactions) are carried out centrally. It is essential to meet the conditions of the system and the constraints caused by it. Ning Zhang proposed an econometric and statistical model to analyze the behavior of generator supply in the NYISO wholesale electricity market one day ahead [15]. If an effective policy can be implemented by NYISO to move generators from a high price bracket to a cheaper price bracket, the effect is likely to be long-lasting. Therefore, in the long term, the market price can be lowered [15].

A rule-based bidding strategy to address various challenges and represent individual market participants has been presented in the form of an agent-based market simulation model [16]. Meanwhile, after comparing power and revenue in different weathers, different markets, and different bidding strategies shown in [17] their proposes a bidding strategy to optimize the bidding behavior of traditional coal-fired units and provides a reference for the bidding strategy of thermal power units when the unified market and renewable energy is connected to the grid in the future. The economic modeling and operation of energy hubs considering the energy market demand and prices have been studied [18] and the most profitable strategies from the electricity operator's perspective in the energy grid have been discussed in depth. On the other hand, related to these conditions, plants that are at high risk of uncertainty in energy output, such as photovoltaic and wind turbines using energy storage, must observe this condition very carefully [19].

In research conducted by [20], there is a minimum income arrangement from power plants using swap techniques related to the production costs of each generating unit, where if more such offers are present in the market, their interactions could open the possibility of strategic bidding. An alternative to conventional multi-unit pay-as-clear type auctions commonly used for electric power exchange clearing was proposed by [21], and some of its characteristic features were analyzed and compared with conventional clearing. Xiang Gao et al. propose a model offering for an integrated PV power generation battery energy storage system (BESS) in the pool-based power market, where the uncertainty of PV generation output is considered [22]. This model considers the market clearing process as an external environment, and through communication with the environmental market, each agent updates the bid price to maximize its income. In the multi-zone integrated energy reserves market model, bidders may submit bids in the form of hourly incremental bids and block bids, which are settled and paid at market-clearing prices (MCP) [23].

The restructuring of the electric power industry aims to eliminate monopolies in the generation and trade sectors, thereby introducing competition at various levels as much as possible [24]. Concepts of the power market have been clearly described by [25–27]. The concept of deregulation and structuring in the power industry is a core of market success. The industry concept in the competitive electricity market has been clearly explained by [28]. A major concern with restructuring the electricity market is the possibility that fuel price volatility could leak into electricity prices [29]. Market power detection techniques were successfully studied by [25]. Factors that give rise to market forces supporting effects that affect consumers and producers must be based on a theoretical and quantitative basis [26]. A study on California's electricity markets breaks market power problems through arbitrage [27].

The definition of pricing was comprehensively studied by [30], especially regarding fuel cost. Implementation of the pricing with the Locational Marginal Pricing (LMP) concept has been formulated by [31–35]. An overview of LMP for the deregulation industry was described in [31]. The study in [32] presented a new market indicated by the market power index based on transmission security constraints. In [33], the main challenge in forecasting LMP is estimating prices accurately in the market day ahead. Mohammad Amin Mirzaei studied a model that integrates the energy market clearing process with the rail route problem [34], in which the space-time network is used to describe the limitations of the rail transport network (RTN). This model is used to determine optimal hourly locations, schedule battery-based energy storage transport system (BEST) charge/discharge, unit thermal power delivery, flexible load scheduling, and find LMP without ignoring the thermal unit daily carbon emission limit. Efforts have been made to determine LMP through the Direct Current-Optimal Power Flow (DC-OPF) model approach by considering losses [35]. In contrast, the determination of the LMP can be used to consider the optimal location of the generator based on the LMP [36].

Practically, the difference in LMP values between locations is due to the consideration of voltage drop, line capacity, losses, etc. The electric energy price is charged to consume not only the LMP but also the cost of transmission losses (TL). This paper proposes calculating energy prices on the consumer side in each location, which is reasonable and fair. The methodology used is a quadratic function approach from the price offered by the supplier so that optimization can be carried out directly [37]. Thus, the determination of the winner of the competition can be decided appropriately. In contrast, the TL is calculated separately through the B-loss matrix approach. Then, allocations of the TL to locations are estimated using the proportional method. In addition to applying several methods, this paper also contributes to formulating the concept of the locational consumer price (LCP) in a competition system. The formulations were tested with a three-location system and six suppliers in 24 auctions with satisfactory results.

In [38], problem-solving was described through distributed coordination, which was applied to a distributed generator to increase energy utilization between the network and the internet. In the power system, the locations formed are connected in an electrical circuit that cannot be separated from each other. Calculation of the optimal power flow and losses must be performed in an integrated manner. Therefore, the algorithm developed in this energy auction issue is through a centralized independent service operator (ISO). In this context, ISO informs the optimization results to stakeholders and to the location operators for follow-up.

For electricity participants, the formation of a bidding strategy in an open access environment is one of the important and most challenging tasks to maximize their profits [39]. This paper creates a framework for determining energy prices on the consumer side of each location through a market mechanism. ISO determines the winner based on offers from Generator Company (GENCO) and Distribution Company (DISCO) that meet transmission network constraints. A GENCO who wins is declared committed, and those who lose are declared uncommitted. ISO performs energy price optimization calculations, while at the same time ensuring committed and non-committed GENCOs to serve loads and losses in determining energy prices on the consumer side. This energy pricing process must be fast (less than five minutes) to have sufficient time to operate the power system, especially to run GENCO, which has been declared the winner, where the bidding process is generally one hour ahead. For the calculation process to run quickly, this paper proposes a direct method for optimization calculations. This method is a method without iteration, so the processing time is fast compared to the iteration method (indirect). The determination of transmission losses is carried out using the B-loss matrix approach. The simulation for the GENCO six system takes 0.04 s to process through Fortran language programming with an Asus Core i3 laptop. With a fast calculation process in determining electricity prices on the consumer side, this proposed model can be operated for large-scale systems, for example, a 118 Bus, 54 GENCO system with a CPU time of 0.37 s.

The layout of this paper comprises several sections. The Section 2 overviews the problem formulation, the Section 3 explains the methodology, and the Section 4 discusses the results. The Section 5 is the conclusion.

## 2. Problem Formulation

Fairness in energy prices is the key to a successful electricity market, especially from the consumer side. Therefore, the application of the LMP mechanism in the electric power system can lead to market success.

For energy prices that are fair to every consumer, this paper proposes a model for calculating energy prices from the consumer side called LCP. The methodology is based on the model through a mechanism in Figure 1. An ISO will optimize energy prices based on supply and demand. Transmission loss is calculated through the B-loss matrix approach, where LCP is calculated based on LMP and the price of losses.

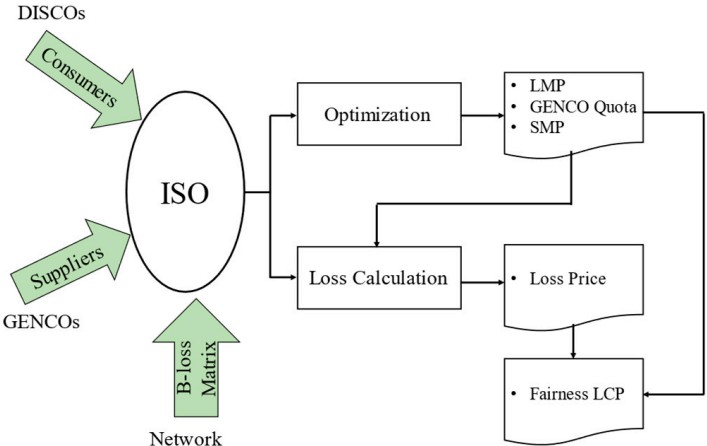

**Figure 1.** LCP Mechanism.

From this explanation, consumers in each location will obtain energy prices based on the energy used and the resulting losses, which in this paper are defined as the energy price on the demand site as follows:

$$\rho_{LP} = \rho_{EP} + \rho_{TP} \tag{1}$$

where $\rho_{LP}$ is the price of electrical energy for consumers in each location, namely, LCP. $\rho_{EP}$ is the price of energy based on LMP. In contrast, $\rho_{TP}$ is the price of transmission loss calculated based on the energy price of the system (marginal price system) and the allocation of transmission losses.

## 3. Methods

### 3.1. Energy Price Optimization

The ISO determines the energy price based on the optimization of the GENCO offers and the amount of power from the DISCO demand in a system based on the mechanism in Figure 1. Generally, the energy offer characteristics of GENCOs can vary, yet in this paper, the offer function in the form of a step function is considered, as shown in Figure 2. This function is a form of function that is not differentiable.

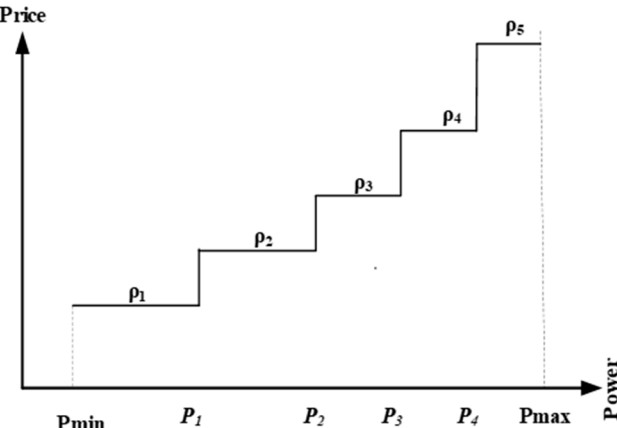

**Figure 2.** Characteristics of the offer function.

The step function in Figure 2 can be expressed by (2).

$$f(P) = \begin{cases} \rho_1, & P_{min} \leq P < P_1 \\ \rho_2, & P_1 \leq P < P_2 \\ \rho_3, & P_2 \leq P < P_3 \\ \rho_4, & P_3 \leq P < P_4 \\ \rho_5, & P_4 \leq P \leq P_{max} \end{cases} \tag{2}$$

where $f(P)$ is a function of fuel price in the form of a step function and $\rho_1$ is the price of fuel in the power range between $P_{min}$ and $P_1$. $P_{min}$ is the minimum power limit, and $P_{max}$ is the maximum power limit.

Therefore, the optimization problem can be formulated with the objective function ($F$) and its constraints as follows:

- Objective Function

$$F = \sum_{i=1}^{n} f_i(P_i) \tag{3}$$

- Equation Constraints

$$P_D = \sum_{i=1}^{n} P_i \tag{4}$$

where $P_D$ is the demand load.
- Inequality Constraints

$$P_{min-i} \leq P_i \leq P_{max-i} \tag{5}$$

As previously mentioned, the paper engaged the direct method suggested by [37]. This method is practical and superior in both speed and accuracy. However, the optimization problem must be in the form of a quadratic function. For this reason, a quadratic approach is used. The characteristics of the offer in Figure 2 can be approximated by a quadratic form, as presented in Figure 3. With this approach, the characteristics of the offers, in general, are shown in (6).

$$f(P) = a + bP + cP^2 \tag{6}$$

where $a$, $b$, and $c$ are parameters of the fuel cost function in the quadratic form.

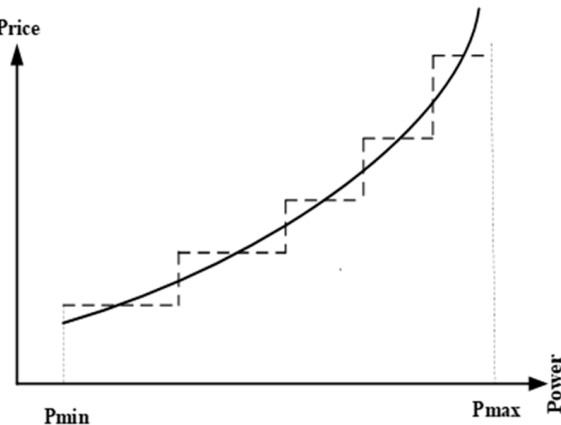

**Figure 3.** Quadratic Approach.

The optimization problem is solved by the LaGrange function, which is:

$$\varnothing = \sum_{i=1}^{n} f_i(P_i) + \lambda \left( \sum_{i=1}^{n} P_i - P_D \right) \tag{7}$$

where $\lambda$ is the LaGrange multiplier. The partial derivative of (7) is

$$\lambda_i = b + 2c_i P_i \tag{8}$$

The optimal conditions must meet the following:

$$\lambda = \lambda_1 = \lambda_2 = \ldots = \lambda_i \tag{9}$$

From (8) and (9), the generator power is obtained, namely

$$P_i = \frac{\lambda - b}{2c_i} \tag{10}$$

For $n$ generators, the power obtained is

$$P_d = \sum_{i=1}^{n} P_i = \sum_{i=1}^{n} \frac{\lambda - b}{2c_i} \tag{11}$$

From (11), the Lagrange multiplier factor can be calculated directly, namely:

$$\lambda = \frac{P_D + \sum_{i=1}^{n} \frac{b_i}{2c_i}}{\sum_{i=1}^{n} \frac{1}{2c_i}} \tag{12}$$

Furthermore, the value of Pi is directly obtained based on the derivative of Equation (6), namely:

$$P_i = \frac{\lambda - b_i}{2a_i} \tag{13}$$

This settlement is still not final and needs to be checked with each GENCO limit with the following conditions:

$$P_i = \frac{\lambda - b_i}{2a_i} \tag{14}$$

a.　　if $P_i > P_{max-i}$, then $P_{opt-i} = P_{max-i}$

b.　　if $P_{min-i} \leq P_i \leq P_{max-i}$, then $P_{opt-i} = P_i$.

c.　　If $P_i < P_{min-i}$, then $P_{opt-i} = 0$.

Note that $P_{opt-i}$ is the optimal power. If conditions a and b occur, then GENCO wins the competition. In contrast, condition c indicates that the competition is lost.

This direct method (DM) has been validated by several methods, namely, the genetic algorithm (GA), lambda iteration (LI), dynamic programming (DP), and large to small area technique (LSAT), for 15 generators. The direct method gives the best results, namely, the lowest cost (32,502.92), as shown in Table 1.

**Table 1.** Comparison of optimization result.

| Gen. | Output Power (MW) | | | | |
|---|---|---|---|---|---|
| | **GA** | **LI** | **DP** | **LSAT** | **DM (Proposed Method)** |
| 1 | 452.40 | 455.00 | 455.00 | 455.00 | 455.00 |
| 2 | 455.00 | 455.00 | 455.00 | 453.00 | 455.00 |
| 3 | 130.96 | 130.00 | 130.00 | 130.00 | 130.00 |
| 4 | 129.10 | 130.00 | 130.00 | 129.60 | 130.00 |
| 5 | 337.10 | 295.30 | 260.00 | 259.70 | 295.30 |
| 6 | 428.50 | 460.00 | 460.00 | 460.00 | 460.00 |
| 7 | 466.40 | 465.00 | 465.00 | 463.60 | 465.00 |
| 8 | 60.00 | 60.00 | 60.00 | 60.00 | 60.00 |
| 9 | 27.60 | 25.00 | 25.00 | 25.00 | 25.00 |
| 10 | 27.10 | 20.00 | 20.00 | 21.90 | 20.00 |
| 11 | 25.70 | 43.40 | 60.00 | 59.00 | 43.37 |
| 12 | 54.00 | 56.30 | 75.00 | 78.20 | 56.33 |
| 13 | 25.00 | 25.00 | 25.00 | 25.00 | 25.00 |
| 14 | 15.00 | 15.00 | 15.00 | 15.00 | 15.00 |
| 15 | 15.00 | 15.00 | 15.00 | 15.00 | 15.00 |
| Total Power | 2648.86 | 2650.00 | 2650.00 | 2650.00 | 2650.00 |
| Cost | 32,517 | 32,503 | 32,506 | 32,507 | 32,502.92 |

The computational time of the DM is much more competitive than that of the iteration method, as the computation time of the DM is 0.22 s and the computation time of the iteration method (e.g., LSAT) is greater than 20 s [40].

### 3.2. Determination of LMP and System Energy Price

The energy prices on GENCO buses are called nodal prices (*NP*). This is determined from the results of optimization, namely:

$$NP_i = b_i + 2c_i P_{opt-i} \tag{15}$$

The *LMP* is defined based on the energy price caused by an increase in demand for one unit of energy and is formulated based on (11).

$$LMP = max\{NP_1, NP_2, \ldots, NP_m\} \tag{16}$$

where *m* is the number of GENCOs at that location. Referring to the LMP definition, the energy price of the system is

$$\rho_{SP} = max\{NP_1, NP_2, \ldots, NP_n\} \tag{17}$$

### 3.3. Determination of Loss Price

Transmission loss is a problem that must be solved in the competition system because transmission losses are a natural property of nature and cannot be neglected. Even though the percentage is small (2–5%), in large electric power systems, it is quite large. For example, 3% of 2000 MW is 60 MW. The calculation of the real transmission losses must be performed by calculating the power flow or the measurement method. Both methods must

be supported by complete data, and the calculation should be fast since the competition time is short (less than 5 min).

This paper calculates transmission losses through an approach using the B-loss matrix. After the generator quotas (as a winner) are obtained from the optimization process, the transmission loss can be calculated through Equation (18).

$$P_{loss} = [P_{opt}][B][P_{opt}] \tag{18}$$

where $B$ is the B-loss matrix. $P_{opt}$ is the optimal power matrix. Meanwhile, the allocation of transmission losses for each location is approached by the proportional method expressed in Equation (19).

$$P_{Li} = \frac{P_{LDi}}{P_D} P_{loss} \tag{19}$$

where $P_{LDi}$ is the number of loads in location $i$. From Equations (17) and (19), the same loss price is obtained at all locations, namely:

$$\rho_{TP} = \frac{P_{loss}}{P_D} \rho_{SP} \tag{20}$$

where $\rho_{TP}$ is the energy price of transmission losses.

## 4. Simulation Results and Discussion

Figure 4 shows a power system used for the numerical simulation to evaluate the proposed method. This system consists of three separate locations. Location-1 contains three GENCOs ($G_1$, $G_3$, and $G_2$) with a total load of DISCOs of 480 MW. Location-2 contains three GENCOs ($G_4$, $G_5$, and $G_6$) with a total load of DISCOs of 430 MW.

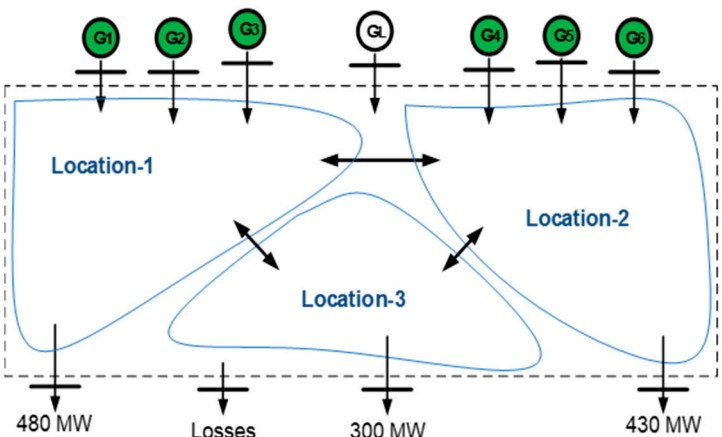

**Figure 4.** The system consisting of three locations.

Location-3 only consists of DISCOs with a total load of 300 MW. In contrast, losses will be supplied by a separated generator $G_L$. Furthermore, Table 2 contains the offer data of GENCOs with four supply blocks ($\alpha$, $\beta$, $\gamma$, and $\pi$). Table 3 assumes the demand for power every hour for 24 h, where the B-loss matrix is presented in (21). The calculation of B-loss matrix components in this paper is not the main discussion. The values of the B-loss matrix components are obtained from separate calculations in the paper.

$$B = 10^{-3} \begin{bmatrix} 1.7 & 1.2 & 0.7 & -0.1 & -0.5 & -0.2 \\ 1.2 & 1.4 & 0.9 & 0.1 & -0.6 & -0.1 \\ 0.7 & 0.9 & 3.1 & 0.0 & -0.1 & -0.6 \\ -0.1 & 0.1 & 0.0 & 0.24 & -0.6 & -0.8 \\ -0.5 & -0.6 & -0.1 & -0.6 & 12.9 & -0.2 \\ -0.2 & -0.1 & -0.6 & -0.8 & -0.2 & 15.0 \end{bmatrix} \tag{21}$$

**Table 2.** Genco offers.

| Genco | α | | β | | γ | | π | |
|---|---|---|---|---|---|---|---|---|
| | Power Level (MW) | Energy Price ($) | Power Level (MW) | Energy Price ($) | Power Level (MW) | Energy Price ($) | Power Level (MW) | Energy Price ($) |
| $G_1$ | 100–225 | 8 | 225–350 | 9 | 350–450 | 9.8 | 450–500 | 10.7 |
| $G_2$ | 50–90 | 6 | 90–135 | 8.2 | 135–175 | 11.9 | 175–200 | 14.2 |
| $G_3$ | 80–150 | 11.2 | 150–215 | 11.7 | 215–260 | 12.5 | 260–300 | 12.7 |
| $G_4$ | 50–80 | 48.5 | 80–110 | 49.6 | 110–135 | 51 | 135–150 | 51.5 |
| $G_5$ | 50–90 | 11.1 | 90–135 | 11.4 | 135–175 | 11.7 | 175–200 | 12 |
| $G_6$ | 80–150 | 52.1 | 150–215 | 52.2 | 215–260 | 52.5 | 260–300 | 53 |

**Table 3.** 24 h Power on Demand Throughput.

| Hour | Demand Load (MW) | Hour | Demand Load (MW) | Hour | Demand Load (MW) | Hour | Demand Load (MW) |
|---|---|---|---|---|---|---|---|
| 1 | 955 | 7 | 989 | 13 | 1220 | 19 | 1159 |
| 2 | 942 | 8 | 1023 | 14 | 1311 | 20 | 1092 |
| 3 | 935 | 9 | 1126 | 15 | 1320 | 21 | 1023 |
| 4 | 930 | 10 | 1180 | 16 | 1350 | 22 | 984 |
| 5 | 935 | 11 | 1198 | 17 | 1321 | 23 | 975 |
| 6 | 965 | 12 | 1210 | 18 | 1262 | 24 | 960 |

The results of optimization analysis at the 12th hour showed that the energy price of the system was $51/MWh. The analysis results are presented in Tables 4–6.

**Table 4.** Result of Quota Analysis of Genco Losses and Nodal Price.

| # | $G_1$ (MW) | $G_2$ (MW) | $G_3$ (MW) | $G_4$ (MW) | $G_5$ (MW) | $G_6$ (MW) | $P_{loss}$ (MW) |
|---|---|---|---|---|---|---|---|
| P | 468.3 | 198.7 | 293.0 | 50.0 | 200.0 | 0.0 | 15.6 |
| NP | 10.7 | 14.2 | 12.7 | 51.5 | 12 | - | - |

**Table 5.** Analysis Result of LMP, Losses, and Customer Price.

| Item | Location-1 | Location-2 | Location-3 |
|---|---|---|---|
| $LMP_{max}$ ($/MWh) | 14.2 | 48.5 | 14.2 |
| Loss (MW) | 6.35 | 5.69 | 3.97 |
| LCP ($/MWh) | 14.881 | 49.181 | 14.881 |

**Table 6.** Power Balance.

| Location | Total Generating Power (MW) | Load (MW) | Power Balance (MW) |
|---|---|---|---|
| Location-1 | 960 | 480 | +495.6 |
| Location-2 | 250 | 430 | −180 |
| Location-3 | 0 | 300 | −300 |
| Losses | 15.6 | 15.6 | 0 |

In this competition, $G_6$ is declared losing, so the quota is zero. This is due to a very high offer, which is above $52/MWh (see Table 2).

For the competition at the 12th hour, Location-1 has a surplus of 495.6 MW, of which 180 MW are exported to Location-2, and the remaining is exported to Location-3. Therefore,

the energy price in Location-3 is determined by the energy price from Location-1. At Location-2, although it imports power from Location-1, the energy price is determined by the LMP itself because the LMP of Location-2 is higher than the LMP of Location-1.

The analysis results for 24-h transactions are listed in Table 7.

**Table 7.** LCP Analysis Results at 3 Location.

| Hour | Location-1 ($/MWh) | Location-2 ($/MWh) | Location-3 ($/MWh) |
|------|--------------------|--------------------|--------------------|
| 1 | 12.865 | 12.865 | 12.865 |
| 2 | 12.661 | 12.661 | 12.661 |
| 3 | 12.659 | 12.659 | 12.659 |
| 4 | 12.659 | 12.659 | 12.659 |
| 5 | 12.659 | 12.659 | 12.659 |
| 6 | 12.866 | 12.866 | 12.866 |
| 7 | 12.875 | 12.875 | 12.875 |
| 8 | 12.869 | 12.869 | 12.869 |
| 9 | 12.865 | 12.865 | 12.865 |
| 10 | 14.367 | 12.867 | 12.867 |
| 11 | 14.836 | 49.136 | 14.836 |
| 12 | 14.881 | 49.181 | 14.881 |
| 13 | 14.824 | 49.324 | 14.824 |
| 14 | 14.856 | 52.756 | 14.856 |
| 15 | 14.855 | 52.755 | 14.855 |
| 16 | 14.860 | 52.760 | 14.860 |
| 17 | 14.855 | 52.755 | 14.855 |
| 18 | 14.834 | 49.134 | 14.834 |
| 19 | 12.866 | 12.866 | 12.866 |
| 20 | 12.873 | 12.873 | 12.873 |
| 21 | 12.873 | 12.873 | 12.873 |
| 22 | 12.869 | 12.869 | 12.869 |
| 23 | 12.868 | 12.868 | 12.868 |
| 24 | 12.865 | 12.865 | 12.865 |

In this simulation, Location-1 always has a surplus so that the excess power is exported to Location-2 and Location-3. This causes the LCP at Location-3 to be the same as the LCP at Location-1, whereas the LCP at Location-2 depends on the LMP itself. If the LMP at Location-2 is lower than the LMP at Location-1, then the LCP at Location-2 is the same as the LCP at Location-1, similar to hours 1–9 and hours 19–24. Beyond this hour, the LCP at Location 2 soared more than 4 times and reached a peak at $52.760/MWh. This was caused by the entry of $G_6$ with a power quota of 92.2 MW, and the price fell to $52.1/MWh (see Table 2).

This method has been tested on a large-scale electrical system, namely 118 buses, 54 generators, based on IEEE 118 Bus data. The simulation results are compared with the calculation results from Newton's Method Power Flow, which are presented in Table 8. Table 8 shows that the results of the proposed method are very close to the results of the Newton method. To supply a load of 4242 MW, the power generated by the proposed method and Newton's method is 4377.59 and 4374.86 MW, respectively.

In other words, the losses generated by the proposed method and Newton's method are 135.59 and 132.86 MW, respectively. Meanwhile, the cost of generating power with the proposed method and Newton's method is $62,556.216 and $62,492.967, respectively. The results of the proposed method have a deviation of 0.0624% for power generation, 2.0548% for losses, and 0.1012% for generation costs from the results of Newton's method.

The large-scale electrical system (IEEE 118 Bus, 54 generators) is divided into five locations, as shown in Table 9. The results of the calculation of energy prices for each location are listed in Table 10. The simulation of this system shows the optimal energy price in each location. The price is determined by the maximum value of all optimized generators at each location (maximum LMP) and their losses. This means that the energy

price at each location will be different, as shown in Table 10. This large-scale simulation requires a very short computation time of 1.28 s.

To realize the energy price at each location in the competition system, for the proposed method, it is fast and guarantees convergence, whereas a new method based on iteration, especially the artificial intelligence method, requires a long computational time to be applied to a large-scale system. In addition, for artificial intelligence methods, such as particle swarm optimization (PSO), there is no guarantee that the solution will fall on the global minimum point, as the PSO is very dependent on the starting point.

**Table 8.** The Comparison Results Between Newton and Proposed Methods.

| Newton's Method | | | | | | Proposed Method | | | | | |
|---|---|---|---|---|---|---|---|---|---|---|---|
| Gen | P (MW) | Gen | P (MW) | Gen | P (MW) | Gen | P (MW) | Gen | P (MW) | Gen | P (MW) |
| 1 | 0 | 19 | 0 | 37 | 477 | 1 | 0 | 19 | 0 | 37 | 477.026 |
| 2 | 0 | 20 | 19 | 38 | 0 | 2 | 0 | 20 | 19.001 | 38 | 0 |
| 3 | 0 | 21 | 204 | 39 | 4 | 3 | 0 | 21 | 204.006 | 39 | 4 |
| 4 | 0 | 22 | 48 | 40 | 607 | 4 | 0 | 22 | 48.002 | 40 | 607.04 |
| 5 | 450 | 23 | 0 | 41 | 0 | 5 | 450.021 | 23 | 0 | 41 | 0 |
| 6 | 85 | 24 | 0 | 42 | 0 | 6 | 85 | 24 | 0 | 42 | 0 |
| 7 | 0 | 25 | 155 | 43 | 0 | 7 | 0 | 25 | 155.006 | 43 | 0 |
| 8 | 0 | 26 | 160 | 44 | 0 | 8 | 0 | 26 | 160.006 | 44 | 0 |
| 9 | 0 | 27 | 0 | 45 | 252 | 9 | 0 | 27 | 0 | 45 | 252.006 |
| 10 | 0 | 28 | 391 | 46 | 40 | 10 | 0 | 28 | 391.021 | 46 | 40.001 |
| 11 | 220 | 29 | 392 | 47 | 0 | 11 | 220.006 | 29 | 392.018 | 47 | 36.001 |
| 12 | 314 | 30 | 513.86 | 48 | 0 | 12 | 314.013 | 30 | 516.415 | 48 | 0 |
| 13 | 0 | 31 | 0 | 49 | 0 | 13 | 0 | 31 | 0 | 49 | 0 |
| 14 | 7 | 32 | 0 | 50 | 0 | 14 | 7 | 32 | 0 | 50 | 0 |
| 15 | 0 | 33 | 0 | 51 | 36 | 15 | 0 | 33 | 0 | 51 | 0 |
| 16 | 0 | 34 | 0 | 52 | 0 | 16 | 0 | 34 | 0 | 52 | 0 |
| 17 | 0 | 35 | 0 | 53 | 0 | 17 | 0 | 35 | 0 | 53 | 0 |
| 18 | 0 | 36 | 0 | 54 | 0 | 18 | 0 | 36 | 0 | 54 | 0 |
| Total Power (MW) | | | 4374.86 | | | | | | 4377.59 | | |
| Total Cost ($) | | | 62,492.967 | | | | | | 62,556.216 | | |
| Losses (MW) | | | 132.86 | | | | | | 135.59 | | |

**Table 9.** Location grouping.

| Location | Bus | Total Demand (MW) |
|---|---|---|
| 1 | 1; 2; 3; 4; 5; 6; 7; 8; 9; 10; 11; 12; 13; 16; 27; 28; 29; 31; 114 | 587 |
| 2 | 14; 15; 17; 18; 19; 20; 21; 22; 23; 25; 26; 30;32; 113; 115; 117 | 376 |
| 3 | 33; 34; 35; 36; 37; 38; 39; 40; 41; 42; 43; 44; 45; 46; 47; 48; 49; 50; 51; 52; 53; 54; 55; 56; 57; 58; 59; 60; 61; 63; 64 | 1342 |
| 4 | 24; 62; 65; 66; 67; 68; 69; 70; 71; 72; 73; 74; 75; 76; 77; 78; 79; 80; 81; 97; 98; 99; 116; 118 | 1033 |
| 5 | 82; 83; 84; 85; 86; 87; 88; 89; 90; 91; 92; 93; 94; 95; 96; 100; 101; 102; 103; 104; 105; 106; 107; 108; 109; 110; 111; 112 | 904 |

**Table 10.** LMP, Losses, and Consumer Prices for a Large-Scale System.

| Location | Load (MW) | LMP$_{max}$ ($/MWH) | Losses (MW) | LCP ($/MWH) |
|---|---|---|---|---|
| 1 | 587.000 | 23.286 | 18.762 | 24.030 |
| 2 | 376.000 | 22.351 | 12.018 | 23.066 |
| 3 | 1342.000 | 23.512 | 42.894 | 24.264 |
| 4 | 1033.000 | 22.373 | 33.018 | 23.088 |
| 5 | 904.000 | 22.450 | 28.894 | 23.168 |

**5. Conclusions**

The LMP calculation has an impact on the realignment of electric power that will be included in the competition mechanism, thus justice from losses can achieve the value of consumer satisfaction in the LCPs. The methods applied must accommodate the GENCO offers and the schedule (timeline) set by the ISO, which uses an auction mechanism one hour ahead. This paper has proposed a fair LCP calculation through the optimization methodology and transmission loss using the B-loss matrix approach. The characteristics of the GENCO offering in the form of step functions are approximated by quadratic functions so that optimization using the direct method can be applied. Transmission loss allocation is based on a proportional approach that is decent enough, where losses are supplied by separated special generators. The energy price of transmission losses is taken as the maximum value of the nodal price of the system. This method was tested through simulations by auctioning 24 times with an electric power system consisting of three locations and 6 GENCOs with satisfactory results. The LCP is very dominantly determined by the LMP rather than the transmission loss price, which is only less than 3%. Location-3 does not have GENCO, so the LCP is determined by the location of the power surplus, namely, Location-1. In Location-2, LCP increased sharply (more than four times) between hours 11–18 because very expensive offers of the GENCOs ($G_4$ and $G_6$) were operating and reached a peak of 52.760 $/MWh, which outside of these hours averaged 12.8 $/MWh. This analysis indicates a positive signal for investors to build cheaper power plants in Location-2 to compete with the participants of $G_4$ and $G_6$.

**Author Contributions:** Conceptualization, Problem Formulation, LCP, and LMP Analysis were done by J.R., and B-Loss Matrix was done by H.Z. All authors have read and agreed to the published version of the manuscript.

**Funding:** This research was funded by Directorate of Research and Community Service, Telkom University and State of Polytechnic of Bandung.

**Institutional Review Board Statement:** Not applicable.

**Informed Consent Statement:** Not applicable.

**Data Availability Statement:** Not applicable.

**Acknowledgments:** This research was funded by the Directorate of Research and Community Service, Telkom University.

**Conflicts of Interest:** The authors declare no conflict of interest.

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
