# Peer review of "Energy Bidding Quadratic Model and the Use of the B-Loss Matrix for Determining Consumer Energy Price"

_applsci, doi:10.3390/app12199743_

Round 1

Reviewer 1 Report

This paper proposes a method for determining electrical energy prices on the consumer side in each location. The method uses a quadratic ap- proach to perform direct method optimization. The transmission losses are calculated through the B-loss matrix approach, and then allocations of the transmission losses are separated with the pro- portional method. Simulation results for three locations with six suppliers, as well as on a larger scale (118 buses, 54 generators) were obtained. The present approach is interesting and necessary for the present research. Overall, I  recommend his publication after some minor points are clarified:

1)     The English is not good enough for publication and needs more polishing.

2)     why the energy price of 352 transmission losses is taken as the maximum value of the nodal price of the system?

3)     This direct method (DM) has been validated by several methods, namely, the genetic 230 algorithm (GA), lambda iteration (LI), dynamic programming (DP), and large to small 231 area technique (LSAT), for 15 generators. why the direct method gives the best results, (Table 1).

Author Response

Original Manuscript ID: applsci-1925894

Original Article Title: “Energy Bidding Quadratic Model and the use of B-loss Matrix for Determining Consumer Energy Price”

Re: Response to reviewer 1

Dear Editor and Reviewers,

Thank you for your suggestions to improve our manuscript.

We are uploading (a) our point-by-point response to the comments (below) (response to reviewers), (b) an updated manuscript with yellow highlighting indicating changes, and (c) a clean updated manuscript without highlights.

Best regards,

<Jangkung Raharjo> et al.

1). Concern # 1: The English is not good enough for publication and needs more polishing.

Author response:  We have improved the English in our manuscripts, so we have changed some words/sentences.

Author action: There are some word/sentence changes in the manuscript

2) Concern # 2: why the energy price of 352 transmission losses is taken as the maximum value of the nodal price of the system?

Author response:  In this paper, it is assumed that the price of losses is in accordance with the energy price of the highest power plant so that the guarantee to cover the energy price can be ensured so that it does not harm the generator that supplies the losses.

Author action: ---

3). Concern # 3:  This direct method (DM) has been validated by several methods, namely, the genetic 230 algorithm (GA), lambda iteration (LI), dynamic programming (DP), and large to small 231 area technique (LSAT), for 15 generators. why the direct method gives the best results, (Table 1).

Author response:  The direct method is one of the optimization methods based on calculus with no approximation. So, the results are guaranteed to be accurate. However, the direct method requires conditions that must be met, namely only for quadratic functions. The accuracy of this method is the same as that of the calculus method based on Lagrange.

Author action: ---

Reviewer 2 Report

The paper present an optimal, fast and accurate approach to determine energy prices for consumers using a quadratic approach implemented through direct method optimization, and the transmission losses are computed using the B-loss matrix approach. 

The research work is of high relevance considering the high energy crises and inflation we are currently facing. However, I have the following concerns and suggestions:

The use of English needs to be improved as the grammar, organization, punctuation, structure, lexicon used made it difficult to read as well as comprehend the manuscript quickly which might dissuade potential readers.

The introduction need to be thoroughly revised as line 26 -33 are typical curating errors that need to be removed and addressed.  

The introduction needs a lot of work as the sentence and line of thought are not that coherent.

line 56 what do you mean by competition? , kindly clarify, and what was the essence of the word.

line 81, What do you mean by contain contain energy storage, wind turbines and photovoltaic (PV),? what do you mean by energy storage?.

Line 121, the word DC-OPF needs to be defined before using it.

Line 148,  GENCO and DISCO abbreviation should be well defined likewise

Line 167 -169 the sentence needs to be rephrased as the sentence has tautology. 

Figure 1 labelling needs to be corrected. (Supplayers to Supplier , Fairnees to Fairness)

Line 178-179  The greek letter (rho) should be replaced with the appropriate one. 

Line 225 the range should vary between pmin_i and p_max_i

Line 40 the nodal prices abbreviation should be written properly Nodal Prices(NP)

Line 246 is redundant and it can be removed as psp was already defined in line 245

Line 286 The NP in the caption for the table should be written in full.

Line 278- The author mentioned the B-matrix was gotten from separate calculation of the paper, can the author shed more light on this, especially giving leads to readers on how to determine the B-loss matrix as well as give relevant references.

Line 307, the prices in table 7 are only similar for hours 1-9 and 19-24, as at hour 10 location 1 and location 2 had different values. 

Author Response

Original Manuscript ID: applsci-1925894

Original Article Title: “Energy Bidding Quadratic Model and the use of B-loss Matrix for Determining Consumer Energy Price”

Re: Response to reviewer 2

Dear Editor and Reviewers,

Thank you for your suggestions to improve our manuscript.

We are uploading (a) our point-by-point response to the comments (below) (response to reviewers), (b) an updated manuscript with yellow highlighting indicating changes, and (c) a clean updated manuscript without highlights.

Best regards,

<Jangkung Raharjo> et al.

1). Concern # 1: The use of English needs to be improved as the grammar, organization, punctuation, structure, and lexicon used made it difficult to read as well as comprehend the manuscript quickly which might dissuade potential readers

Author response:  We have improved the English in our manuscripts, so we have changed some words/sentences.

Author action: There are some word/sentence changes in the manuscript

2) Concern # 2: The introduction needs to be thoroughly revised as line 26 -33 are typical curating errors that need to be removed and addressed

Author response:  We apologize, the first paragraph of the introduction (lines 26-34) has typical curation errors. And we have removed it.

Author action: We have removed the first paragraph (lines 26-34) in the introduction, which is highlighted in red.

3). Concern # 3:  The introduction needs a lot of work as the sentence and line of thought are not that coherent.

Author response:  we have tried to improve the introduction, for that we made some changes to the paragraphs in the manuscript.

Author action: There are sentence/paragraph changes in the manuscript.

4). Concern # 4. line 56 what do you mean by competition? kindly clarify, and what was the essence of the word.

Author response:  Competition is an Electric re-structurization based on a market mechanism, where independent participants (competitors), both generating companies and distribution companies will compete through auctions and electrical energy bids managed by independent operators.

Author action: ---

5). Concern # 5. line 81, What do you mean by contain energy storage, wind turbines and photovoltaic (PV),? what do you mean by energy storage?.

Author response:  Wind turbines and PV are energy generators that have intermittent characteristics, so they must be handled properly, for example with energy storage. Energy storage is a device for storing electrical energy (such as a battery), this device needs to be provided for alternative generators due to intermittent and non-constant energy supply. Where this device is useful for stabilizing the energy supply.

Author action: ---

6). Concern # 6. Line 121, the word DC-OPF needs to be defined before using it.

Author response:  DC-OPF is Direct Current-Optimal Power Flow

Author action: We've added it to the manuscript

7). Concern # 7. Line 148,  GENCO and DISCO abbreviation should be well defined likewise

Author response:  GENCO stands for Generator Company. Generator Company is an independent generator company as a supplier of electrical energy in the competition system.

DISCO stands for Distribution Company is an independent distribution company as a buyer of electrical energy in a competition system.

Author action:  We've added it to the manuscript

8). Concern # 8. Line 167 -169 the sentence needs to be rephrased as the sentence has tautology. 

Author response:  We have rearranged the sentences on lines 167-169 and added some words on lines 170.

Author action: Changes in sentences and words in lines 167-170, are marked in yellow

9). Concern # 9. Figure 1 labelling needs to be corrected. (Supplayers to Supplier , Fairnees to Fairness)

Author response:  We have made a typo. We have made a typo. The correct ones should be "supplier" and "Fairness".

Author action: Figure 1 in the manuscript has been corrected

10). Concern # 10: Line 178-179  The greek letter (rho) should be replaced with the appropriate one.

Author response:  We've done a typo. We have corrected the manuscripts we marked in yellow.

Author action: We have corrected the manuscripts we marked in yellow.

11). Concern # 11: Line 225 the range should vary between pmin_i and p_max_i

Author response:  We've made a typo on line 225 and we've corrected it.

Author action: We've corrected it by changing the Pi to Pmax-i.

12). Concern # 12: Line 40 the nodal prices abbreviation should be written properly Nodal Prices (NP)

Author response: We've made a typo "..called nodal (NP) Prices", which should be "...called nodal prices (NP)"

Author action: We've corrected the manuscript with a yellow mark

13). Concern # 13: Line 246 is redundant and it can be removed as psp was already defined in line 245

Author response: We have removed "where  is the energy price of the system"

Author action: We removed "where  is the energy price of the system", which is marked in red in the manuscript

14). Concern # 14: Line 286 The NP in the caption for the table should be written in full.

Author response: We have changed the "NP" on line 286 to "Nodal Price".

Author action: We've corrected the manuscript with a yellow mark.

15). Concern # 15: Line 278- The author mentioned the B-matrix was gotten from separate calculations of the paper, can the author shed more light on this, especially giving leads to readers on how to determine the B-loss matrix as well as give relevant references.

Author response:  The calculation of the B loss matrix refers to:

  1. Bin Liu, Feng Liu, Wei Wei, Jingran Wang, Estimating B-coefficients of power loss formula considering volatile power injections: an enhanced least square approach, IET Generation, Transmission & Distribution, 2018, Vol. 12 Iss. 12, pp. 2854-2860.
  2. Allen J. Wood, Power Generation, Operation, and Control, John Wiley & Sons, Inc., 1984, pp. 116-120.

Author action: ---

16). Concern # 16: Line 307, the prices in table 7 are only similar for hours 1-9 and 19-24, as at hour 10 location 1 and location 2 had different values

Author response:  We've made a typo on line 307. It was written "...1-10", it should be "...1-9".

Author action: We've corrected the manuscript with a yellow mark.
